# Assessment of Concentration KRT6 Proteins in Tumor and Matching Surgical Margin from Patients with Head and Neck Squamous Cell Carcinoma

**DOI:** 10.3390/ijms25137356

**Published:** 2024-07-04

**Authors:** Dariusz Nałęcz, Agata Świętek, Dorota Hudy, Karol Wiczkowski, Zofia Złotopolska, Joanna Katarzyna Strzelczyk

**Affiliations:** 1Department of Otolaryngology and Maxillofacial Surgery, St. Vincent De Paul Hospital, 1 Wójta Radtkego St., 81-348 Gdynia, Poland; zzlotopolska@gmail.com; 2Department of Medical and Molecular Biology, Faculty of Medical Sciences in Zabrze, Medical University of Silesia in Katowice, 19 Jordana St., 41-808 Zabrze, Polandkarolwicz@gmail.com (K.W.); jstrzelczyk@sum.edu.pl (J.K.S.); 3Silesia LabMed Research and Implementation Centre, Medical University of Silesia in Katowice, 19 Jordana St., 41-808 Zabrze, Poland; 4Students’ Scientific Association, Department of Medical and Molecular Biology, Medical University of Silesia, Katowice, 19 Jordana St., 41-808 Zabrze, Poland

**Keywords:** HNSCC, cancerogenesis, protein level, tumor, surgical margin, KRT6A, KRT6B, KRT6C, cytokeratins, KRT6s

## Abstract

Head and neck squamous cell carcinomas (HNSCCs) are one of the most frequently detected cancers in the world; not all mechanisms related to the expression of keratin in this type of cancer are known. The aim of this study was to evaluate type II cytokeratins (KRT): KRT6A, KRT6B, and KRT6C protein concentrations in 54 tumor and margin samples of head and neck squamous cell carcinoma (HNSCC). Moreover, we examined a possible association between protein concentration and the clinical and demographic variables. Protein concentrations were measured using enzyme-linked immunosorbent assay (ELISA). Significantly higher KRT6A protein concentration was found in HNSCC samples compared to surgical margins. An inverse relationship was observed for KRT6B and KRT6C proteins. We showed an association between the KRT6C protein level and clinical parameters T and N in tumor and margin samples. When analyzing the effect of smoking and drinking on KRT6A, KRT6B, and KRT6C levels, we demonstrated a statistically significant difference between regular or occasional tobacco and alcohol habits and patients who do not have any tobacco and alcohol habits in tumor and margin samples. Moreover, we found an association between KRT6B and KRT6C concentration and proliferative index Ki-67 and HPV status in tumor samples. Our results showed that concentrations of KRT6s were different in the tumor and the margin samples and varied in relation to clinical and demographic parameters. We add information to the current knowledge about the role of KRT6s isoforms in HNSCC. We speculate that variations in the studied isoforms of the KRT6 protein could be due to the presence and development of the tumor and its microenvironment. It is important to note that the analyses were performed in tumor and surgical margins and can provide more accurate information on the function in normal and cancer cells and regulation in response to various factors.

## 1. Introduction

Head and neck cancers (HNSCC) are significant clinical and social problems [1]. Their share of all malignant neoplasms in Poland has invariably ranged between 5.5 and 6.2% in recent years, which accounts for approximately 5500 to 6000 new cases each year [2,3]. They are characterized by an overall poor prognosis, and the long-term overall survival (5 years) in this group of patients is about 50% for all stages of HNSCC [4]. The causes of head and neck cancer are not completely clear, but it is known that cigarette smokers and alcohol abusers are at risk of the disease [5,6]. Several molecular mechanisms are known to be involved in the development and metastasis of cancer [6,7,8,9]. Some studies indicate that type II cytokeratins may also influence the process of carcinogenesis.

KRT6 (keratin 6) protein is a member of type II cytokeratins, of which three isoforms are known (KRT6A, KRT6B, and KRT6C), encoded by three genes: *KRT6A*, *KRT6B*, and *KRT6C*, respectively. They are found in the epidermis of the palms and soles, the filiform papillae of the tongue, the epithelium lining the mucosa of the mouth and esophagus, the epithelial cells of the nail bed, and hair follicles [10]. Recent studies have demonstrated the likely involvement of keratins in the processes of carcinogenesis, including cancer cell invasion and the formation of metastasis [11,12]. Therefore, these molecules have found their application as diagnostic markers for cancer, especially in unclear clinical cases, which is particularly valuable for correct identification of the tumor and selection of the most appropriate treatments [13,14,15,16,17]. The relationship between changes in the concentration of KRT6s proteins and HNSCC is still unknown and appears to have dual roles as both protooncogenes and tumor suppressors [13,17,18,19,20]. The role of KRT6 isoforms could be associated with the type of tumors. To the best of our understanding, it remains unclear whether alterations in KRT6s isoform concentrations have any association with HNSCC. Due to the role of KRT6 proteins in cell growth, invasion, and migration processes, it is hypothesized that the levels of KRT6A, KRT6B, and KRT6C will be changed in tumor samples compared to margin samples in patients with HNSCC.

The aim of this study was to evaluate the levels of selected proteins in tumors and the matched samples of surgical margins, in a group of patients with primary HNSCC. The association of clinical-pathological and demographical variables with the concentration of the proteins studied was also analyzed.

## 2. Results

### 2.1. Concentration of the Selected KRTs in Tumour Samples and Margin

Significantly higher KRT6A protein concentration was found in the tumor samples than in the margin sample (0.01976 (0.00962–0.02825) vs. 0.00933 (0.00488–0.01938)); (*p* = 0.0003). However, in the analysis of KRT6B and KRT6C, we observed significantly higher concentration of protein in the margin than in the tumor specimens (for KRT6B 1.264 (0.63545–1.80653) vs. 0.518 (0.215–0.8901)); (*p* = 0.0009) and (for KRT6C 0.15724 (0.10418–0.30146) vs. 0.13441 (0.079336–0.18031)); (*p* = 0.0274). The results are given in Figure 1.

### 2.2. Protein Level of KRT6 and Cancer Classification, and Localization of Tumour

In the case of cancer classification due to a small number of some types only two groups were tested—OSCC (30 cases) and the combined group consisted of OPSCC, LSCC, and HPSCC subtypes (20 combined cases). There were no observed significant differences in KRT6 protein levels between those groups. The most of primary tumor regions used in this study were localized in the larynx (17 samples, 31.48%), the floor of the mouth (14, 25.93%), the tongue (10, 18.52%), the jaw (5, 9.26%). There were no observed differences in localization in KRT6 protein concentration. In both cases, there were no observed differences in the association of groups with sex, clinical parameters, tobacco, and alcohol habits (Appendix A).

### 2.3. Protein Level of KRT6 and Clinical Parameters

In the case of the nodal status, a combined group of patients with nodal status N2 and N3 was formed, due to a small number of N3 patients. The higher concentrations of KRT6B in margin samples showed a group of patients with higher nodal status, which was statistically significant (N0 vs. N2 + N3; 0.83449 (0.48671–1.5046) vs. 1.61342 (1.277–2.3565)); (*p* = 0.0200). The results are shown in Figure 2A.

KRT6C protein concentration was significantly higher in the margin samples in the group of patients with nodal status N2 or N3, as compared to samples in the group of patients with nodal status N0 (0.29846 (0.23041–0.34369) vs. 0.12591 (0.07801–0.20545)); (*p* = 0.0052). There was a significant difference in age between the groups of patients with N0 and N2 + N3 status, with higher nodal status related to younger age (*p* = 0.01958; 66.77 ± 11.75 vs. 56.17 ± 9.32). Tables with characteristics with clinical parameter groups are presented in Appendix A. There were no observed associations with other parameters (T and G). The linkage of group T1 with T2, because of a small number of cases in the T1 group, did not obtain significant results vs. T3 and T4 cases as individual groups or joined groups.

### 2.4. Protein Level of KRT6 and Tobacco and Alcohol Habits

In the group of smokers, compared to non-smokers, KRT6A protein concentration was higher in the tumor (0.02126 (0.01055–0.03044) vs. 0.00858 (0.00669–0.01089)); (*p* = 0.0061). An opposite situation was observed for KRT6B in margins in the group of smokers compared to non-smokers (1.048 (0.59183–1.54050) vs. 2.357 (1.45890–4.5156)); (*p* = 0.0168). These results are shown in Figure 3A,B. There was no observed correlation between KRT6 proteins and the amount of cigarettes per day or with years of smoking but there was a medium positive correlation of KRT6B protein in tumor samples with pack-years (0.32; *p* = 0.0407).

Furthermore, regular alcohol drinkers had higher levels of KRT6B, compared to occasional drinkers 0.88750 (0.47487–1.78950) vs. 0.32989 (0.16330–0.62059); (*p* = 0.0047) in the tumor. Similar observations were found in the same groups in the surgical margin (1.49696 (0.98943–3.34760) vs. 1.05949 (0.44043–1.59470)); (*p* = 0.0272). The results are given in Figure 3C. Significantly higher KRT6C protein level was found in the margin samples collected from regular drinkers, in comparison to the abstinent patients (0.24470 (0.15152–0.30080) vs. 0.05722 (0.04014–0.10430)); (*p* = 0.0059) (Figure 3D). The group that was regularly drinking was significantly younger than the abstinent group (*p* = 0.0069; 53 (51–65) vs. 79 (71.5–82.75)) but this could be the effect of group size. Tables with characteristics of the tobacco and alcohol habits groups are presented in Appendix A. No association with other parameters was observed.

### 2.5. Protein Level of KRT6 and Proliferation Index Evaluated by Ki-67

The patients with proliferation index Ki-67 > 20 had a significantly higher KRT6C concentration than patients with Ki-67 ≤ 20 (0.13789 (0.08662–0.17819) vs. 0.05961 (0.04652–0.1062)); (*p* = 0.0267) in the tumor samples (Figure 4). A table with characteristics of the proliferative index evaluated ≤ 20 and Ki-67 > 20 is presented in Appendix A. No association with other parameters was observed.

### 2.6. KRT6 Protein Concentration and HPV Status and p16 Status

The median protein concentration of KRT6B in the tumor was higher in HPV(+) patients than in the group of HPV(−) (2.43711 (0.59399–4.02520) vs. 0.36460 (0.16355–0.60257)); (*p* = 0.0335). Similarly, HPV(+) patients had a higher median level of KRT6C in the tumor samples (0.19478 (0.18870–0.26409) vs. 0.10846 (0.07466–0.16088)); (*p* = 0.0199), which is presented in Figure 5A,B.

The concentration of KRT6B protein was higher in the tumor samples of the p16(+) group than in the p16(−) group (2.43711 (0.59399–4.0252) vs. 0.31389 (0.17382–0.62325)); (*p* = 0.0327). Also in the tumor samples, the KRT6C protein had a higher level in the p16(+) group than in p16(−) (0.19478 (0.18870–0.26409) vs. 0.10560 (0.06749–0.17158)); (*p* = 0.0451). Results are presented in Figure 5CD. Tables with characteristics of the p16 and HPV groups are presented in Appendix A. For HPV groups there was no association with other parameters observed (Appendix A). For the p16 groups there was an association with the smoking status (*p* = 0.014), group p16(−) included more patients who were smokers (Appendix A).

### 2.7. Correlation of KRT6 Proteins

KRT6C protein from the margin samples was significantly mildly correlating with all three KRT6 proteins in the tumor samples. With KRT6A, a negative correlation was observed, presented in Table 1 in blue (−0.34; *p* = 0.0156), and with KRT6B (0.39; *p* = 0.01193) and KRT6C (0.35; *p* = 0.01458) positive correlations were observed as presented in orange in Table 1. KRT6C showed also a stronger correlation (0.53; *p* = 0.000195) with KRT6B in the margin samples; a positive correlation was presented in Table 1 in orange. 

## 3. Discussion

The exact role of KRT6A, KRT6B, and KRT6C in HNSCC is still unclear. Based on our knowledge (databases PubMed and Medline), this has been the first study to analyze the concentration of these proteins by ELISA in the tumor and margin samples, obtained from patients with HNSCC.

In our study, the median KRT6A protein level was significantly higher in the HNSCC tumor sample than in the margin. In HNSCC and OSCC in other types of KRT (KRT17, KRT19), similarly to ours, increased protein levels were observed in the tumor compared to the healthy sample or surgical margin [21,22,23]. Some studies found upregulated KRT6A levels in other types of the cancer sample, compared to matched normal samples [13,24,25,26]. The high KRT6A concentration was reported in the analysis of non-small-cell lung cancer (NSCLC) and lung adenocarcinoma (LADC) and is associated with lymph node metastasis and advanced T stage cancer [13,24,25]. In addition, the authors reported that KRT6A overexpression can affect the upregulation of G6PD (glucose-6-phosphate dehydrogenase), resulting in activation of the metabolic pathway promoting invasion and the growth of cancer cells [25]. Additionally, the increased KRT6A expression was observed in the nasopharyngeal carcinoma cell line, and it was noted that KRT6A silencing was associated with inhibition of cell invasion and metastasis formation via the β-catenin pathway [18]. Also, Chen et al. suggested that KRT6A played a prominent role in promoting proliferation and adhesion, with simultaneous inhibiting tumor cell apoptosis in bladder cancer [27]. In contrast, a study on the immune microenvironment in pancreatic ductal adenocarcinoma showed that KRT6A could modulate the function of tumor-associated macrophages (TAMs), an important part of the leukocyte infiltrate, through other proteins and molecular pathways [28]. In addition, evidence suggests that keratin proteins may be involved in migration, adhesion, cell proliferation, and regulation of keratinocyte inflammation, and play an important physiological role in cell repair [29]. Therefore, based on our observation, we suggested that KRT6A could play an important role in HNSCC, however, in order to confirm this hypothesis, it would be important to conduct studies on a larger group of patients than ours.

On the other hand, in our study, lower levels of KRT6B and KRT6C proteins were observed in HNSCC samples, compared to the resected surgical margin. In the case of other types of KRT (KRT13, KRT14, KRT24) in HNSCC, OSCC, and neoplastic oral mucosa, decreased protein levels were observed in the tumor sample compared to the margin/healthy tissue, which is consistent with our results [30,31,32,33,34]. Similar to our results, the in silico breast cancer studies found reduced KRT6B and KRT6C levels in the tumor tissues, compared to the cancer-free samples [20,35,36]. On the contrary, increased KRT6B expression was demonstrated in colorectal cancer, compared to healthy colorectal mucosal tissue, based on the results of bioinformatics analyses [16]. Higher levels of KRT6C in saliva were observed in OSCC compared to the control group [37]. A recent study confirmed that OSCC tumor margin cells had unique transcriptomic profiles and ligand–receptor interactions [38,39]. In addition, it is suspected that KRT6B may be involved in the process of immune response, including M2 polarization of macrophages [40]. The role of KRT6B and KRT6C proteins in cancerogenesis is unclear. KRT protein levels may vary according to different types of cancer, which could be associated with specific molecular characteristics of the tumor as well as the demographic, clinical, and pathological parameters [34]. Moreover, the cited studies may have used different protein detection methods, varying in sensitivity and specificity, which resulted in different KRT level results obtained [20,34,35,36,37,38,39]. The increased levels of KRT6B and KRT6C proteins in the tumor margin observed in our study might therefore indicate that these proteins may affect the immune response and the immune microenvironment.

In our research, KRT6B and KRT6C protein concentrations in the margin tissue samples in the group of patients with the nodal status N2 or N3 were significantly higher, as compared to samples in the group of patients with the nodal status N0. In a study evaluating the salivary proteins as potential biomarkers for the early diagnosis of OSCC, it was reported that KRT6C was significantly elevated at the tumor stage T3/T4, compared to T1/T2, whereas in our study we did not note any differences [37]. Importantly, Liu et al. and Song et al. detected elevated KRT6B levels in the bladder tumors, which correlated positively with the metastatic status and the stage of the disease [40,41]. Other authors reported the elevated KRT6C levels in lung adenocarcinoma cell lines to be associated with cell proliferation, migration, and invasion [42]. Our study demonstrated the elevated levels of these proteins, which may be related to their functions in the epithelial-mesenchymal transition—an important process for epithelial cells to achieve invasiveness, whereas it would be important to repeat the research on a larger number of samples.

Our study reported that smokers showed significantly increased concentrations of KRT6A in the tumor samples and significantly decreased concentrations of KRT6B in the margin tissue. Importantly, it has been shown that exposure to cigarette smoke is able to inhibit DNA repair, immunosuppression, induction of oxidative stress, and induction changes in the proteome of oral keranocytes, resulting in both, upregulation and downregulation of selected proteins [43,44]. The authors suggest that the consequences of smoking may therefore be oral cancerous lesions [45]. The studies based on proteomic analysis of exhaled breath condensate in the group of patients with lung cancer, in the healthy volunteers and additionally on primary human gingival epithelial cells, showed that KRT6A and KRT6B proteins were elevated in 58.1% of smokers in both groups [46,47]. The study based on bioinformatic analyses in NSCLC showed that the increased KRT6A expression was associated with current and past smoking habits and KRT6As in NSCLC function as oncogenes and may be useful as potential prognostic diagnostic biomarkers of NSCLC in smokers [48]. In another study, the authors suspect that keratins are proteins that respond to oxidative stress, and the impaired expression of KRT may be a response to disruption of the oral mucosal barrier exposed to tobacco smoke [49]. Based on our results and observations, the effects of cigarette smoke may take place through modulation of the expression of cytokeratin type II protein in response to epithelial damage. It is possible that additional genetic and epigenetic mechanisms are present in smokers that may be involved in the altered expression of KRT family genes and proteins.

Moreover, we reported the increased concentration of KRT6B protein in the tumor and the margin samples of regular alcohol drinkers compared to occasional drinkers. A higher KRT6C protein level was noted in the margin samples collected from regular drinkers, in comparison to the abstinent patients. Subsequent publications confirm alcohol drinking as a risk factor in malignancies of the oral cavity, pharynx, and larynx [50,51,52,53,54]. As one potential factor, non-pathogenic strains colonizing the oral cavity have been shown to be responsible for the processes that convert ethanol to acetaldehyde, which is able to modify the processes of methylation, synthesis, and DNA repair, and could interact with proteins, which ultimately implies cell damage and proliferation. In addition, the authors also reported the ability of acetaldehyde to activate the oncogenic transcription factors in oral keratinocytes [51,52,53,54]. Therefore, we suggested that the upregulation of KRT6B and KRT6C protein levels in the tumor and margin samples of regular alcohol drinkers, compared to occasional drinkers, could be related to increased keratinization, in response to the tissue damage caused by alcohol and its metabolites.

In this study, we observed that tumor samples with proliferation index Ki-67 > 20 showed significantly higher KRT6C concentration than those with Ki-67 ≤ 20. The Ki-67 protein is one of the most commonly used markers due to its correlation with tumor proliferation. Various studies have shown higher Ki-67 expression in OSCC tissues than in normal tissues and increased with the progression of dysplasia in oral mucosa tissues [55,56,57]. Keratins are the main structural proteins of the epithelial cells, so we suspect that the increased level of KRT6C may be related to the role of these proteins in proliferation and repair processes in response to disorders in the epithelial cells.

Our study showed that the median protein concentrations of KRT6B and KRT6C were higher in HPV(+) patients than in the group of HPV(−) in tumor tissues. Moreover, we found a similar correlation in the case of p16 status, where we obtained higher levels of KRT6B and KRT6C proteins in p16-positive tumor samples. In the study by Woods et al., similar to ours, increased levels of KRT7 and KRT19 were observed in HPV-positive OPSCC compared to HPV-negative cases, which may indicate the involvement of keratins in the etiopathogenesis of HPV-related OPSCC [58]. Studies have shown that HPV is associated with infection of the basal layer of the epithelium, and then to complete the life cycle of the virus; it uses the pathways of epithelial proliferation and differentiation into keratinocytes [59,60]. The study by Zhang et al. showed that HPV(+) HNSCC tumors are characterized by increased expression of genes related to keratinocyte differentiation processes [60]. In an in vitro studies, dysregulated interferon signaling, DNA replication, and DNA damage response pathways were observed in HPV(+) keranocyte cells [61,62]. In addition, in cervical cancer, it was reported that the presence of HPV16, and thus the effect of E6/E7 oncoproteins, was associated with increased expression of keratinization genes [63]. As it is known, HNSCC is a heterogeneous disease entity due to HPV infection status, so we suggest that there are differences in oncogenic pathways, probably in addition to E6 and E7 oncoproteins, related to HPV genes associated with various processes that can modulate the expression of a large number of genes and proteins, including KRT6B and KRT6C, which in response can result in increased keratinization.

## 4. Materials and Methods

### 4.1. Study Population

The study group consisted of 54 patients (66.67% male and 33.33% female) diagnosed with HNSCC. Specimens of the tumor and of the corresponding margins were collected after surgical resection at the Department of Otolaryngology and Maxillary Surgery, St. Vincent De Paul Hospital, Gdynia, Poland. The collected samples were examined histologically and classified as primary HNSCC. Marginal samples were taken from the surgical margin at least 10 mm from the tumor border and were histologically confirmed as being free of cancer. Classification and staging of tumor specimens were performed according to the 8th edition of the AJCC Cancer Staging Manual [64]. All the resected specimens were secured and then transported in dry ice to the Laboratory of Medical and Molecular Biology, Medical University of Silesia in Katowice, Poland, where they were stored at −80 °C until further analysis. The HNSCC comprised 30 cases of oral squamous cell carcinoma (OSCC), 2 cases of oropharyngeal squamous cell carcinoma (OPSCC), 17 cases of laryngeal squamous cell carcinoma (LSCC), 2 cases of hypopharyngeal squamous cell carcinoma (HPSCC), 2 cases of nasal cavity squamous cell carcinoma (NCSCC), and 1 case of skin squamous cell carcinoma (SSCC). Primary tumor regions used in this study were localized in the larynx (17 samples, 31.48%), the floor of the mouth (14, 25.93%), the tongue (10, 18.52%), the jaw (5, 9.26%), the oral cavity (2, 3.70%), the retromolar trigone (4, 7.40%), and one each in the cheek (1, 1.85%), and the soft palate (1, 1.85%). The main inclusion criteria for the HNSCC group included a diagnosis of a primary tumor and no preoperative radio/chemotherapy. All patients participating in the study gave written informed consent. Approval for the study was obtained from the Bioethical Committee, Regional Medical Chamber in Gdansk (no. KB-42/21). Table 2 shows the characteristics of the study group.

Smokers were using tobacco for 32.61 ± 11.83 pack-years on average. Drinking status was determined by patient surveys (abstinent, occasionally) and information about alcohol abuse or alcoholism disease (regularly).

### 4.2. Homogenization and Total Protein Concentration

First, 10% tissue homogenates were obtained in PRO 200 mechanical homogenizer (PRO Scientific Inc., Oxford, CT, USA) at the rate of 10,000 rpm, in the presence of an appropriate volume of cooled PBS buffer (Eurx, Gdansk, Poland). The commercial AccuOrange™ Protein Quantitation Kit (Biotium, Fremont, CA, USA) was used to quantify the total protein. The determinations were carried out in the previously prepared tissue homogenates, in duplicate, according to the manufacturer’s instructions, without dilutions. The detection range of the assay was 0.1–15 µg/mL of protein. Fluorescence was measured at the excitation wavelength of 480 nm and the emission wavelength of 598 nm (SYNERGY H1 microplate reader; BIOTEK, Winooski, VT, USA using the Gen5 2.06 software). Total protein determinations were performed to express the KRT6A, KRT6B, and KRT6C concentrations in units per µg of total protein.

The homogenates were dissected and then frozen at −80 °C until further analysis.

### 4.3. Enzyme-Linked Immunosorbent Assay (ELISA) Kits for Proteins Concentration

The protein levels of KRT6A, KRT6B, and KRT6C were assayed in homogenates by the ELISA method, according to the standard instruction (ELISA Kits Cloud-Clone Corp., Katy, TX, USA); (Assay ID: SED234Hu for KRT6A, SEA486Hu for KRT6B and SED758Hu for KRT6C). To determine the concentrations of the tested samples, a calibration curve was prepared using the standards included in the kit. Absorbance readings were recorded at 450 nm wavelength and calibrated according to the standard curve in ng/mL (SYNERGY H1 microplate reader; BIOTEK, Winooski, VT, USA). The tests were characterized by the following sensitivities: 6.1 pg/mL for KRT6A, 0.060 ng/mL for KRT6B, and 13.3 pg/mL for KRT6C. The intra-assay variation was below 10% and the inter-assays were below 12% for all the determined proteins. All standards and samples were run in duplicate. The results obtained were calculated corresponding to the total protein concentration and presented in pg/µg.

### 4.4. DNA Isolation and HPV Detection

The tissue samples were homogenized with Lysing Matrix A (MP Biomedicals, Irvine, CA, USA). DNA was isolated with a GeneMATRIX Tissue DNA Purification Kit (Eurx, Gdansk, Poland) according to the protocol. The quality and quantity of isolated DNA were assessed on a spectrophotometer (NanoPhotometer Pearl, Implen, Munich, Germany) and DNA was kept at −20 °C until HPV analysis.

HPV status was assessed with GeneFlow^TM^ HPV Array Test Kit (DiagCor Bioscience Ltd., Kowloon Bay, Hongkong) with FT^PRO^ Flow-through System (DiagCor Bioscience Ltd., Kowloon Bay, Hongkong) and FT^PRO^ Auto System (DiagCor Bioscience Ltd., Kowloon Bay, Hongkong) according to instruction. First, isolated DNA was used in PCR reaction on Mastercycler Personal Thermal Cycler (Eppendorf, Hamburg, Germany). Next PCR products were denatured and hybridized, and after enzyme conjugation and color development, the results were scanned with FT^PRO^ Auto System (DiagCor Bioscience Ltd., Kowloon Bay, Hongkong). Positive and negative controls provided by the manufacturer were performed in all runs.

### 4.5. Evaluation of p16 and Ki-67 by Immunohistochemical Staining

Immunohistochemical analysis for p16 was performed by commercial kit according to the manufacturer’s instructions (CINtec p16 Histology, Roche MTM Laboratories, Mannheim, Germany). The staining made use of BenchMark ULTRA automated system (Roche Diagnostics, Basel, Switzerland). The immunohistochemical analysis for Ki-67 was evaluated using the CONFIRM anti-Ki-67 (30-9) Rabbit Monoclonal Primary Antibody (Ventana Medical Systems, Inc., Tucson, AZ, USA) using BenchMark ULTRA in automatic mode (Roche Diagnostics, Basel, Switzerland). The expression of proliferation index Ki-67 was categorized into 2 groups: Ki-67 > 20 and Ki-67 ≤ 20.

p16 and Ki-67 expression status were evaluated following the instructions provided in the kit package.

### 4.6. Statistical Analysis

The results from ELISA and the patient surveys were tested with Shapiro–Wilk to determine normality for groups with less than or equal to 5 cases non-normal distribution was assumed. Differences between the groups and the tissue type were tested with the Mann–Whitney U test or Kruskal–Wallis with Dunn–Sidak post hoc. Data in Appendix A were processed with the chi^2^ test with Bonferroni correction for multiple comparisons should more than one comparison be carried out. Differences in age were tested with the Students *t*-test or One-Way ANOVA with Tukey HSD post hoc or Mann–Whitney U test or Kruskal–Wallis with Dunn–Sidak in case of small groups (≤5 cases). Correlations were performed with the Spearman rank coefficient method. Significant results were indicated with *p* ≤ 0.05 or *p* ≤ 0.05/(number of comparisons). All calculations were performed with Statistica 13.1 (TIBCO Software Inc., Palo Alto, CA, USA) or with Excel 2019 (Microsoft, Redmond, WA, USA) software. Significant data are presented as box plots with the median in the middle and the 1st and 3rd quartile as a box with minimum and maximum values as whiskers. Data mentioned in the text are presented as median with the 1st and 3rd quartile as follows, median (quartile 1st–quartile 3rd).

## 5. Conclusions

Our results showed that concentrations of KRT6s were different in the tumor and the margins samples and varied in relation to clinical and demographic parameters. We add information to the current knowledge about the role of KRT6s isoforms in HNSCC. Moreover, our study identified the changes in concentration of selected KRT proteins in both, tumor and surgical margin samples from HNSCC patients, and can provide more accurate information on the function in normal and cancer cells and regulation in response to various factors. Observation of the levels of KRT6 isoforms in the tumor and the margin shows that the isoforms play different roles and functions in diverse cell types which also depends on the HPV/p16 status, the proliferative index Ki-67, and the tobacco and alcohol habits. There is also an association between the levels of these proteins and clinical and pathological variables, such as tumor clinical stage and nodal status, which could be due to the presence and development of the tumor and its microenvironment.

The main limitation of the study was the small size of the samples. Therefore, to confirm the associations more accurately, future studies should be conducted on larger cohorts. In addition, tests with cell lines and animal models are required, which could provide valuable information for a better understanding of the role in tumorgenesis of KRT6s proteins. Furthermore, considering the analyzed cytokeratins type II proteins are needed to understand fully their impact on cancer prognosis and progression. Our future studies will then focus on the analysis of the disease-free and the overall survival in HNSCC patients.

## Figures and Tables

**Figure 1 ijms-25-07356-f001:**
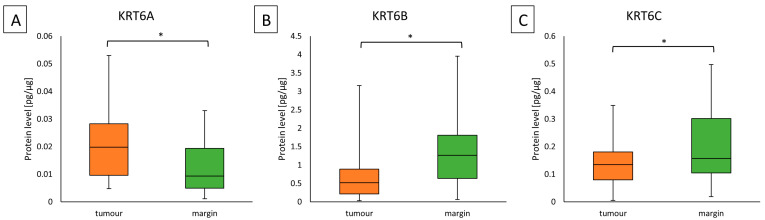
Results of the KRT6 proteins level analyses in tumor samples compared to margin samples. (**A**)—Protein KRT6A in tumor and margin samples, (**B**)—Protein KRT6B in tumor and margin samples, (**C**)—Protein KRT6C in tumor and margin samples. Statistical analysis was carried out with the Mann–Whitney U test where differences with *p* ≤ 0.05 were considered statistically significant. The orange blocks indicate the tumor while the green ones represent the margin sample. * Indicate significant differences.

**Figure 2 ijms-25-07356-f002:**
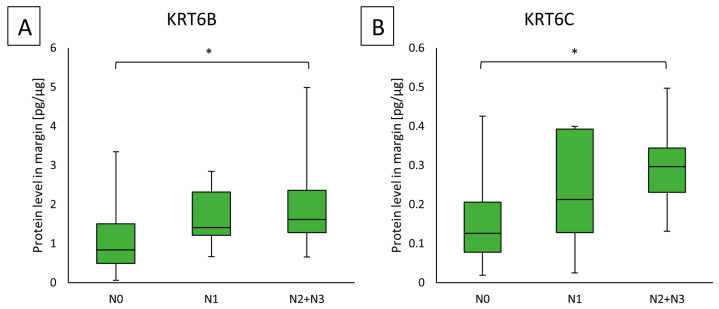
(**A**) The KRT6B protein level in the margin sample samples in the group of patients with N0, N1, and N2 + N3 nodal status; (**B**) the KRT6C protein level in the margin samples in the group of patients with N0, N1, and N2 + N3 nodal status. Statistical analysis was performed with Kruskal–Wallis and with Dunn–Sidak post hoc, differences of *p* ≤ 0.05 are considered statistically significant. The orange blocks indicate the tumor while the green ones represent the margin sample. * Indicate significant differences.

**Figure 3 ijms-25-07356-f003:**
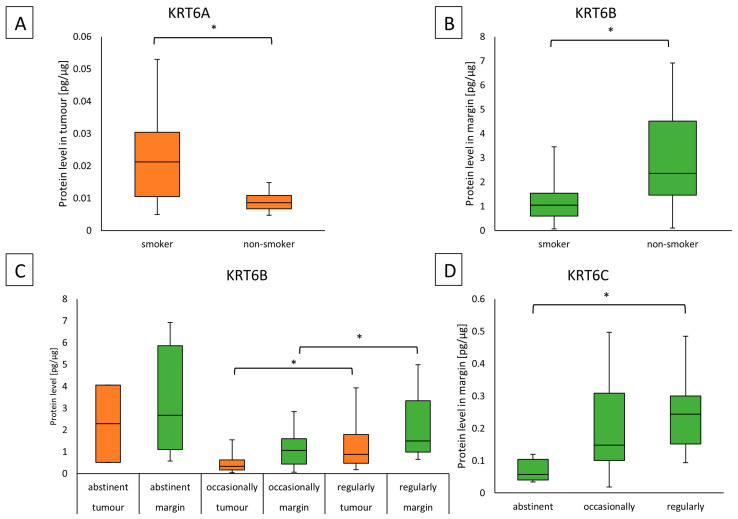
(**A**) The KRT6A protein level in the tumor samples according to smoking status; (**B**) the KRT6B protein level in the margin samples according to the smoking status; (**C**) the KRT6B protein level in the tumor and margin samples according to the drinking status; (**D**) the KRT6C protein level in the margin samples according to the drinking status. Statistical analysis for AB charts was performed with the Mann–Whitney U test, for CD charts Kruskal–Wallis was performed with Dunn–Sidak post hoc and differences with *p* ≤ 0.05 are considered as statistically significant results. The orange blocks indicate the tumor while the green ones represent the margin sample. * Indicate significant differences.

**Figure 4 ijms-25-07356-f004:**
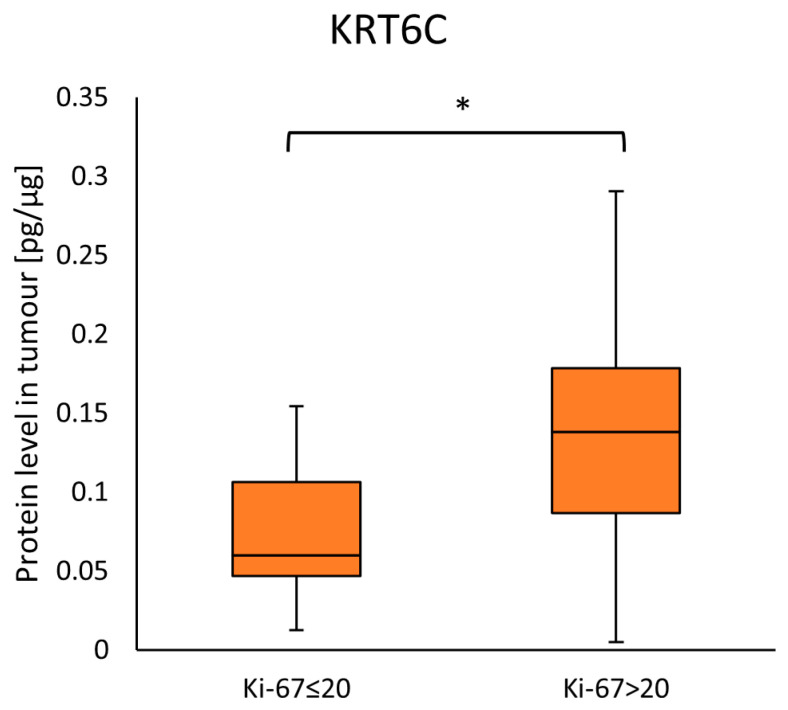
The KRT6C protein level in the tumor samples according to Ki-67 protein. Statistical analysis was performed with the Mann–Whitney U test and differences with *p* ≤ 0.05 are considered statistically significant. The orange blocks indicate the tumor. * Indicate significant difference.

**Figure 5 ijms-25-07356-f005:**
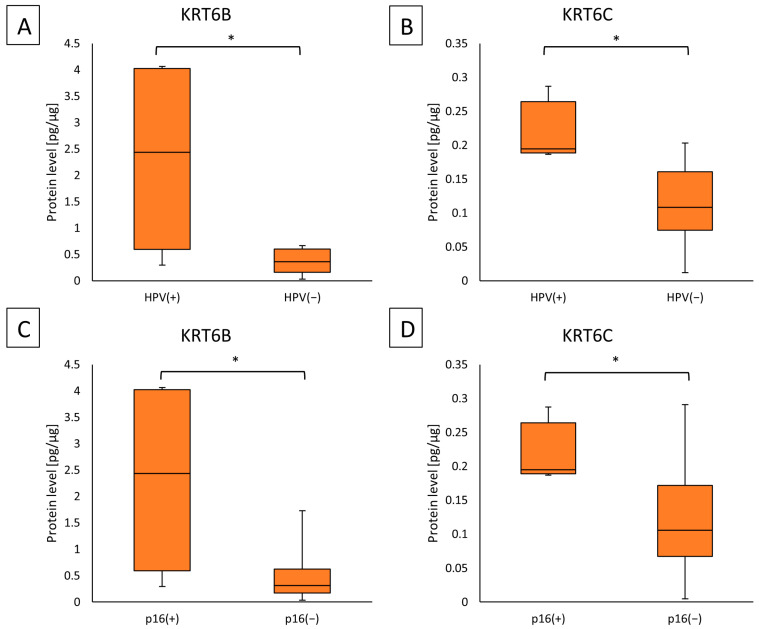
(**A**) The KRT6B protein level in the tumor samples according to the HPV status; (**B**) the KRT6C protein level in the tumor samples according to the HPV status; (**C**) the KRT6B protein level in the tumor samples according to p16 status; (**D**) the KRT6C protein level in the tumor samples according to p16 status. Statistical analysis was performed with the Mann–Whitney U test and differences with *p* ≤ 0.05 are considered statistically significant. The orange blocks indicate the tumor. * Indicate significant differences.

**Table 1 ijms-25-07356-t001:** *p*-value and R^2^ of Spearman correlation for KRT6 proteins.

*p*-Value		Tumor			Margin		
		KRT6A	KRT6B	KRT6C	KRT6A	KRT6B	KRT6C
tumor	KRT6A	1.00	0.45	0.23	0.48	0.16	**0.02**
	KRT6B	0.45	1.00	0.15	0.37	0.36	**0.01**
	KRT6C	0.23	0.15	1.00	0.94	0.37	**0.01**
margin	KRT6A	0.48	0.37	0.94	1.00	0.48	0.86
	KRT6B	0.16	0.36	0.37	0.48	1.00	**0.00**
	KRT6C	**0.02**	**0.01**	**0.01**	0.86	**0.00**	1.00
**Spearman’s Rank** **Coefficient**		**Tumor**			**Margin**		
		**KRT6A**	**KRT6B**	**KRT6C**	**KRT6A**	**KRT6B**	**KRT6C**
tumor	KRT6A	1.00	0.12	−0.18	−0.11	−0.21	−0.34
	KRT6B	0.12	1.00	0.23	0.15	0.16	0.39
	KRT6C	−0.18	0.23	1.00	0.01	−0.14	0.35
margin	KRT6A	−0.11	0.15	0.01	1.00	−0.12	0.03
	KRT6B	−0.21	0.16	−0.14	−0.12	1.00	0.53
	KRT6C	−0.34	0.39	0.35	0.03	0.53	1.00

The bold *p*-value indicates significance. Orange color indicates a positive correlation, and blue indicates a negative correlation.

**Table 2 ijms-25-07356-t002:** Parameters characterizing the HNSCC patients.

Parameter	N (%)
Sex	
female	18 (33.33)
male	36 (66.67)
Average age (range)	64 (53–72)
Smoking	
yes	45 (83.33)
no	9 (16.67)
Drinking	
yes occasionally	36 (66.67)
yes regularly	15 (27.78)
no	3 (5.56)
Clinical T-classification (T)	
T1	3 (5.56)
T2	9 (16.67)
T3	19 (35.19)
T4	20 (37.04)
NA *	3 (5.56)
Nodal status (N)	
N0	31 (57.41)
N1	6 (11.11)
N2	12 (22.22)
N3	2 (3.70)
NA *	3 (5.56)
Metastasis (M)	
M0	52 (96.30)
M1	0 (0.00)
NA *	2 (3.70)
Histological grading (G)	
G1	15 (27.78)
G2	32 (59.26)
G3	6 (11.11)
G4	1 (1.85)

* NA—not assessed.

## Data Availability

The data used to support the findings of this research are available upon request.

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
