# Peer review of "Assessment of Concentration KRT6 Proteins in Tumor and Matching Surgical Margin from Patients with Head and Neck Squamous Cell Carcinoma"

_ijms, 2024, doi:10.3390/ijms25137356_

Round 1
Reviewer 1 Report
Comments and Suggestions for Authors
Assessment of concentration KRT6 proteins in tumor and 2 matching surgical margin from patients with head and neck 3 squamous cell carcinoma
Dear Authors.
Thank you for having investigated the levels of selected Keratin proteins in tumors and in samples of surgical margins, in a group of patients with primary HNSCC.
This is an an interesting study of interest in researchers in this field.
Here my comments.
Abstract :
- KRT should be defined in full.
- stimulant do not seems appropriate word (tobacco and alcohol habits)
- the sentence association with tumor progression and carcinogenesis seems to be speculative
Material and Methods
- The quantity of tobacco and alcohol should be provided in pack per year and unit in order to uniform the sample
Discussion
- The assumption of the role seems to be speculative since the presence of the protein in itself does not provide an explanation on the mechanism of cancer formation or progression. The sentences should be in my opinion amended .
- One limit has been suggested (sample size). You could add also the necessity of studies on cultured cells and in animals to better understand the role in tumorogenesis of these proteins.
Author Response
Assessment of concentration KRT6 proteins in tumor and matching surgical margin from patients with head and neck squamous cell carcinoma”
by Dariusz Nałęcz, Agata Świętek, Dorota Hudy, Karol Wiczkowski, Zofia Złotopolska, Joanna Katarzyna Strzelczyk
We would like to thank the reviewers for their valuable and detailed comments, suggestions and their time spent on reviewing the manuscript. We believe that after completion of the suggested edits, the revised manuscript has improved in the overall presentation and clarity.
To facilitate the work of the reviewers, we attached a marked-up version of the manuscript.
Looking forward hearing from you soon.
Sincerely,
Authors
Reviewer:
Dear Authors.
Thank you for having investigated the levels of selected Keratin proteins in tumors and in samples of surgical margins, in a group of patients with primary HNSCC.
This is an interesting study of interest in researchers in this field.
Here my comments.
Abstract :
- KRT should be defined in full.
Thank you for the suggestion. We have added the full name of the protein family in the abstract.
- stimulant do not seems appropriate word (tobacco and alcohol habits)
Thank you for the comment. We have changed the ,,simulant" in the text of the article to ,,tobacco and alcohol habits".
- the sentence association with tumor progression and carcinogenesis seems to be speculative
Thank you for the suggestion. We have we have amended this part of the text.
Material and Methods
- The quantity of tobacco and alcohol should be provided in pack per year and unit in order to uniform the sample
Thank you for the suggestion. We added information about the mean pack-year for tobacco users, and the correlation of cigarette amount per day, pack-years, and active smoking years with KRT6 proteins was assessed. For alcohol consumption, we didn’t have information about amounts, but we had information about alcoholism disease or alcohol abuse – the group was divided with that information for regular drinkers. Occasional drinkers were people without disease or abuse of alcohol, but still, they mentioned drinking in the survey, and, lastly, abstinent were people who, in the survey, marked no alcohol consumption.
Discussion
- The assumption of the role seems to be speculative since the presence of the protein in itself does not provide an explanation on the mechanism of cancer formation or progression. The sentences should be in my opinion amended .
Thank you for the important suggestion. We have we have amended this part of the text.
- One limit has been suggested (sample size). You could add also the necessity of studies on cultured cells and in animals to better understand the role in tumorogenesis of these proteins.
Thank you for the comment. We have added this information in the study limitation.
Reviewer 2 Report
Comments and Suggestions for Authors
Dear editor
Thank you for the invitation for the review of this article. This is well done work. However, my main concern is the small sample and with some different locations and stage, grade.... When grouping the cases the samples is not so homogeneous as should be. also this gives some limitations to the study. for example differences betwenn T1 and T2. this has no logic as T1 has only 3 patienst, if I understand well. How can be compared the expression of markers between a group with only 3 patients??? I suggest to design the variables regarding this.
Also a limitation of the studis is mandatory in the discussion, adressing for example this problems.
Comments on the Quality of English LanguageDear editor
Thank you for the invitation for the review of this article. This is well done work. However, my main concern is the small sample and with some different locations and stage, grade.... When grouping the cases the samples is not so homogeneous as should be. also this gives some limitations to the study. for example differences betwenn T1 and T2. this has no logic as T1 has only 3 patienst, if I understand well. How can be compared the expression of markers between a group with only 3 patients??? I suggest to design the variables regarding this.
Also a limitation of the studis is mandatory in the discussion, adressing for example this problems.
Author Response
Assessment of concentration KRT6 proteins in tumor and matching surgical margin from patients with head and neck squamous cell carcinoma”
by Dariusz Nałęcz, Agata Świętek, Dorota Hudy, Karol Wiczkowski, Zofia Złotopolska, Joanna Katarzyna Strzelczyk
We would like to thank the reviewers for their valuable and detailed comments, suggestions and their time spent on reviewing the manuscript. We believe that after completion of the suggested edits, the revised manuscript has improved in the overall presentation and clarity.
To facilitate the work of the reviewers, we attached a marked-up version of the manuscript.
Looking forward hearing from you soon.
Sincerely,
Authors
REVIEWER
Dear editor
Thank you for the invitation for the review of this article. This is well done work. However, my main concern is the small sample and with some different locations and stage, grade.... When grouping the cases the samples is not so homogeneous as should be. also this gives some limitations to the study. for example differences betwenn T1 and T2. this has no logic as T1 has only 3 patienst, if I understand well. How can be compared the expression of markers between a group with only 3 patients??? I suggest to design the variables regarding this.
Thank you for your suggestion. We tried to eliminate the T1 group or consider the joint T1+T2 group. Still, in both cases, there was no significant difference between groups with different T statuses, so we corrected this part in the manuscript.
Also a limitation of the studis is mandatory in the discussion, adressing for example this problems.
Thank you for the comment. At the end of the discussion there is information about the limitation of our study, while, as suggested, we have also added in the discussion information about the need to carry out studies on a larger sample group/patients to verify the hypothesis.
Round 2
Reviewer 1 Report
Comments and Suggestions for Authors
Dear Dr Nałęcz,
Thank you for having modified the manuscript according to the comments made.
I believe that in the present form it ca no be accepted for publication.
Best wishes.